# Influence of Wax and Silver Nanoparticles on Preservation Quality of Murcott Mandarin Fruit during Cold Storage and after Shelf-Life

Mohamed M. Gemail [1,†], Ibrahim Eid Elesawi [2,3,*,†], Muthana M. Jghef [4], Badr Alharthi [5], Woroud A. Alsanei [6], Chunli Chen [2], Sayed M. El-Hefnawi [1] and Mohamed M. Gad [1]

1   Horticulture Department, Faculty of Agriculture, Zagazig University, Zagazig 44511, Egypt
2   College of Life Science and Technology, Huazhong Agricultural University, Wuhan 430070, China
3   Agricultural Biochemistry Department, Faculty of Agriculture, Zagazig University, Zagazig 44511, Egypt
4   Department of Radiology, College of Medical Technology, Al-Kitab University, Kirkuk 36001, Iraq
5   Department of Biology, University College of Al Khurmah, Taif University, P.O. Box 11099, Taif 21944, Saudi Arabia
6   Department of Food and Nutrition, Faculty of Human Sciences and Design, King Abdulaziz University, Jeddah 21589, Saudi Arabia
*   Correspondence: ibrahimeid@zu.edu.eg
†   These authors contributed equally to this work.

**Abstract:** Citrus fruits are perishable and considered the most prominent and essential crops at the local and global levels. The world is focused on minimizing fruit postharvest losses, maintaining fruit quality, and prolonging its storability and marketability. Thus, this study was carried out throughout the two successive seasons of 2018 and 2019 on Murcott mandarin fruits, with the purpose of extending their storage period and shelf life by making a mixture of nanosilver and wax as a coating. The fruits were picked on the first of March, washed, and coated with the following treatments: 1000 ppm imazalil (IMZ as a control), wax, 50 ppm nanosilver, 100 ppm nanosilver, and finally, the combination of wax plus 100 ppm nanosilver, packaged in 0.005% perforated polyethylene (PPE), and stored at 5 ± 1 °C and 90%–95% relative humidity for four months. Samples of each treatment were randomly taken at monthly intervals to evaluate the tested treatments' effects on fruit quality during cold storage and 6 days of shelf life. The data proved that the combination of wax plus 100 ppm nanosilver packaged in 0.005% perforated polyethylene (PPE) was the most effective treatment for reducing discarded fruits, fresh weight loss, and catalase enzyme activity, as well as maintaining pulp firmness and vitamin C content and keeping a better taste panel index. Therefore, these coatings could be promising alternative materials for extending mandarin fruits' postharvest life and marketing period.

**Keywords:** mandarin; edible coating; AgNPs; postharvest; quality

## 1. Introduction

Citrus is considered the most prominent and important fruit crop at the local and global levels. Furthermore, in Egypt, citrus is the backbone of the fruit crop due to its most significant economic importance compared with other types of fruit, taking the first rank in the cultivated area as the first export crop. Moreover, it has been a large horticultural industry during the last few years, and the cultivated area has reached about 1941 Km$^2$ and produced 4,323,030 tons [1]. Furthermore, it is considered the most popular fruit in Egypt and has a high nutritional value with a rich content of vitamins, organic acids, pigments (carotenoids, flavonoids, anthocyanin, thiamine, riboflavin, niacin, etc.), sugars, fibers, essential and volatile oils, as well as mineral elements such as calcium, phosphorus, iron, sodium, and potassium [2–5].

Citrus fruits are deemed perishable and susceptible to a reduced quality after harvest due to decay and water loss during transpiration and respiration [6]. Considering that citrus fruits have natural wax on the cortex that gets eliminated through the fruit's prolonged washing process, accordingly, compensation is needed to avoid dehydration [7,8]. The Murcott mandarin is one of the most popular mandarin cultivars in Egypt, but it is also exposed to many losses after harvesting and during storage, leading to a shorter postharvest life [9].

In the past decade, the world has attempted to reduce the loss of crops postharvest and maintain the quality of the fruit during storage and marketing. Inferior handling affects postharvest quality, disease incidence, and sensitivity to a chilling injury and contributes to high postharvest and marketing chain losses and reduces the storage period. Thus, these losses can occur at all postharvest stages until consumption [10–13]. Currently, some researchers aim at reducing excessive chemical components in crop fertilization by inexpensively utilizing environmentally safe organic substances to improve plant quality. Consequently, postharvest practices control fruit damage by using safe, suitable, and efficient harvest, handling, and storage treatments to prolong postharvest shelf life [10,14].

Fruit coating is considered a practical technique to provide additional preservation versus physiological disorders after harvest like stem-end rind breakdown, chilling injury [15,16], and a prolonged storage period, improving fruit appearance and quality [12,17]. Furthermore, experience with papaya, mandarin, and plums shows that edible coatings have the effect of maintaining postharvest fruit quality [18–21]. Furthermore, guar gum can be commercially applied to coat fruit to extend its shelf life and preserve postharvest quality in mango and "Valencia" oranges [22,23]. Wax is considered the most remarkable postharvest implementation to limit unfavorable changes and elongate the shelf life of fruit. The important properties of wax coatings on citrus fruit are a good, lustrous, and appealing appearance, which continues during the marketing process, the reduction of fruit weight loss, and maintenance of fruit quality. Furthermore, wax is predicted to be beneficial as a transporter of fungicides [6,15,24,25].

Moreover, the wax implementation plays a paramount function in increasing fruit quality. Furthermore, the imazalil (IMZ) preserves fruit against green mold caused by *Penicillium digitatum* and a single application of IMZ in wax has controlled green mold well and inhibited sporulation, with differing impacts on many parameters of fruit quality [26]. Over the previous years, new technologies have been introduced to prolong fresh fruit shelf life, such as loading coating substances with nanoparticles, which has presented an innovative and safe fruit-defense mechanism that ensures minimum direct exposition and lower penetration of nanoparticles into the treated food products [27,28].

Nanotechnology has attracted attention in the last decade due to its vital applications in many fields such as medicine, pharmaceuticals, catalysis, materials, and energy [29]. Nanomaterials are used in sustainable agriculture as promising plant growth agents, fertilizers, and pesticides. Moreover, nanomaterials are used in the control of plant pests, including insects, fungi, and weeds [29,30]. Applications of NPs are used in agriculture for a more efficient and safe use of chemicals. Although there are slight effects of toxicity on seed germination and root growth of five higher plant species—radish, rape, lettuce, corn, and cucumber—silver nanoparticles aid seedling growth in wheat, when large amounts are used including alumina, magnetite ($Fe_3O_4$), zinc, and zinc oxide [30–34]. On the contrary, silver nanoparticles can stimulate wheat growth and production and their application in the soil has very promising growth-promoting effects on wheat growth and yield [33]. Many studies have used nanotechnology in the field of food production, where it is preferable to use biosynthetic nanoparticles [33,35].

The research community has the most interest in silver nanoparticles (AgNPs) due to their noteworthy properties in size and effective antibacterial activities [36,37]. Silver nanoparticles have been used as food additives and packaging materials to eliminate pathogens [27,38,39]. Additionally, edible coating formulations mixed with AgNPs can be applied as a palatable fruit coating to reduce the growth of microorganisms that cause

postharvest diseases to increase their shelf life [40–42]. Furthermore, the treatment with AgNPs exhibited significantly decreased weight losses compared with uncoated orange fruits [43]. Adding AgNPs to the polyethylene significantly reduces weight loss, retards softening, prevents fruit corruption, reduces decay, maintains firmness, decreases the rapid reduction in citric acid and vitamin C contents, and increases total antioxidant activity in fruit [44–46]. The studies also showed an increase in fruit weight retention, rate of respiration, total sugars, total soluble solids, and total carotenoids through the storage period. In contrast, this increase was relatively minimal and significant in coated fruits compared with uncoated fruits. On the other hand, hardness and acidity are greatly reduced upon storage. Still, this reduction was low for coated fruits when AgNPs and carboxymethyl cellulose CMC-AgNP coatings were capable of retardation fruit ripening of mango and preserving fruit quality through a cold storage period [41,47–51].

Nevertheless, there are no issued data on the use of coating substances loaded with AgNPs for improving mandarin fruit behavior during storage, especially the fruit of the Murcott mandarin cultivar. Furthermore, the studies that concentrated on the attitude of this cultivar and its quality characteristics over cold storage and shelf life are few [52–56]. Therefore, this work aims to study the influence of coating with wax and different concentrations of AgNPs before packaging in perforated polyethylene on the postharvest storage behavior and quality attributes of a Murcott mandarin fruit cultivar during its cold storage and shelf-life period.

## 2. Materials and Methods

### 2.1. Fruit Material and Growth Conditions

This study was performed during two successive seasons, 2018 and 2019, on mature yellow mandarin (*Citrus reticulata* L. Osbeck) fruits cv. Murcott. Fruits (600 fruits, 150 fruits/treatment, 10 fruits/replicate) were selected from a private citrus orchard in Wadi El-Molak, Ismailia Region Governorate, Egypt. The trees were six years old, budded on Volkameriana lemon rootstock, grown in sandy loam soil 5 m apart, and received the standard horticultural practices adopted in the area. The fruits were collected on the first day of March in both seasons. The collected fruits were uniform, healthy, and as free of physiological disorders and visible pathological problems as possible. All fruits were carefully transferred to the postharvest laboratory in the Horticulture Department, Faculty of Agriculture, Zagazig University, and kept for 24 h at room temperature. After that, all fruits were washed completely with tap water and soap and quickly rinsed with water to remove soap residues. Then, the fruits were surface sterilized with a 0.5% solution of sodium ortho-phenylphenate (SOPP) at pH 11.8–12.1 and 32 °C for two minutes, then left to air dry before treatment. Then the fruits were dipped in the treatment solution for two minutes and then left to air dry. The treatments included coating with 1000 ppm IMZ (control), wax, 50 and, 100 ppm nanosilver, and finally, the combination of wax plus 100 ppm nanosilver. Coated mandarin fruits of all treatments were air dried and packaged in 0.005% perforated polyethylene (PPE) and stored at 5 ± 1 °C and 90%–95% relative humidity for four months. Samples of each treatment were randomly taken at monthly intervals to evaluate the effect of the tested treatment during cold storage and six days' shelf life. The silver nanoparticles were obtained from Ahmed Saad's lab [57], which used phenolic aqueous extracts from pomegranate and watermelon peels to convert silver nitrate to silver nanoparticles, which were described as stated in the report [57].

### 2.2. Fruit Quality Characteristics

#### 2.2.1. Discarded Fruit

This parameter was calculated as the percentage of rejected fruits to the total fruits [58].

#### 2.2.2. Fruit Weight Loss (FWL)

To estimate FWL at specific storage periods, the fruits of each replicate were separately weighed before and after treatment during the cold storage process and before and after

six days of shelf life. *FWL* was calculated as a percentage of the initial weight according to [47,59] utilizing the following Equation (1):

$$FWL\ (\%) = ((Wi - Ws)/W1) \times 100. \tag{1}$$

where (*Wi*) is fruit weight at the initial period, and (*Ws*) is fruit weight at the sampling period.

### 2.2.3. Fruit Pulp Firmness (FPF)

Five fruits from each replicate were hand flaked and used to distinguish pulp firmness as $g/cm^2$ using a Push-Pull Dynamometer (Model FD 101) [58,60].

### 2.2.4. Juice Total Soluble Solid Percentage (TSS)

TSS percentage was determined using a hand refractometer as Brix°.

### 2.2.5. Juice Total Acidity Percentage (TA %)

TA percentage was estimated by titrating 0.1 N NaOH in the presence of phenolphthalein as an indicator, and the result was calculated as grams of citric acid per 100 mL of fruit juice using, according to A.O.A.C. [61].

### 2.2.6. Ascorbic Acid (Vitamin C) Content

The content of ascorbic acid was estimated by titration in 2,6 dichlorophenol-indophenol dye, estimated and expressed as milligrams per 100 mL of juice [62].

### 2.2.7. Juice Panel Test Index (PTI)

Five persons judged random fruit samples from each replicate to give PTI scores according to the following index: Excellent taste = 4; Very good taste = 3; Good taste = 2; Acceptable taste = 1, and Bad taste (unacceptable fruits) = 0 [63].

### 2.3. Enzymes and Antioxidant Determination

### 2.3.1. Preparation of the Extract

The mandarin peels were ground after drying in a vacuum oven (45 °C), and 10 g of each powder sample was taken and then homogenized in 100 mL of 50% ethanol (1:10, *w/v*) stirred for 3 h at room temperature. The samples were filtered to obtain the supernatant. Then the solvent was disposed of with a rotary evaporator [64].

### 2.3.2. DPPH Radical Scavenging Assay

The total antioxidant activities in the mandarin peel extracts with different treatments at a concentration of (500 μg/mL) were examined according to [65]. We mixed 100 μL of each solution with 1 μL of ethanolic DPPH in the microtiter plate wells and incubated it at room temperature in the dark for 30 min. A microtiter plate reader (BioTek Elx808, USA) was used to measure the absorbance at 517 nm b and then applied Equation (2).

$$Radical\ scavenging\ activity\ (\%) = \frac{(Abs.\ control - Abs.\ sample)}{(Abs.\ control)} \times 100 \tag{2}$$

### 2.3.3. Catalase Activity (CAT)

CAT activity was assayed by using Biodiagnostic, Kit No. CA 25 17, Egypt, according to the method described by [66,67]. The formed chromophore absorbance was inversely proportional to the amount of catalase in the experimented sample [68]. Briefly, we mixed 0.05 mL of the sample, 0.5 mL of phosphate buffer (pH = 7), and 0.1 mL of chromogen-inhibitor and incubated for one min at room temperature, added 0.50 mL $H_2O_2$ and 0.20 mL chromogen-inhibitor to the mixture then incubated for 10 min at 37 °C. The decrease in absorbance was recorded at 510 nm.

*2.4. Statistical Analysis*

Before running a one-way ANOVA, pretests were conducted. We tested the normality assumption on sample distributions and obtained *p*-values of 0.0001. for homogeneity; we used the Levene test with *p*-value = 0.01598. The triplicate data means were analyzed for statistical differences by one-way ANOVA at a confidence level of 95% [69], using Costat program version 6.4 (Costat 2008). The sample size was calculated using the following Equation (3)

$$n = (\frac{ZSD}{E})^2 \tag{3}$$

Means were compared with the least significant difference (LSD) as a post hoc test at a probability level of 5%.

## 3. Results

*3.1. Discarded Fruit %*

After harvest, citrus fruits are considered perishable and vulnerable to quality decline due to rot and water loss from transpiration and respiration [6]. The data referred to the influence of various postharvest treatments on discarded fruit percentage of Murcott mandarins, regardless of the cold storage period, as illustrated in (Table 1).

**Table 1.** Effect of postharvest treatments on discarded fruit percentage (DFP%) of Murcott mandarin fruits during 1, 2, 3, and 4 months of cold storage and after 6 days of shelf life during 2018 and 2019 seasons.

| Treatments | Cold Storage Period (Month) (P) | | | | | 6 Days Life (P) | | | | |
|---|---|---|---|---|---|---|---|---|---|---|
| | The First Season (2018) | | | | | | | | | |
| | 1 | 2 | 3 | 4 | Mean | 1 | 2 | 3 | 4 | Mean |
| IMZ (Control) | 0.00 e | 3.32 cd | 6.71 b | 10.15 a | 5.04 A | 0.00 f | 6.70 bc | 10.03 a | 10.20 a | 6.73 A |
| Wax | 0.00 e | 0.00 e | 3.40 cd | 6.66 b | 2.51 B | 0.00 f | 0.00 f | 6.42 bc | 6.83 b | 3.31 B |
| not wax + 50 ppm nanosilver | 0.00 e | 0.00 e | 3.34 cd | 3.50 c | 1.71 C | 0.00 f | 0.00 f | 6.52bc | 6.62 c | 3.20 B |
| not wax + 100 ppm nanosilver | 0.00 e | 0.00 e | 3.28 d | 3.39 cd | 1.67 C | 0.00 f | 0.00 f | 3.63 e | 6.29 bc | 2.01 C |
| wax + 100 ppm nanosilver | 0.00 e | 0.00 e | 0.00 e | 3.41 cd | 0.85 D | 0.00 f | 0.00 f | 0.00 f | 4.40 d | 1.66 D |
| Mean | 0.00 D | 0.66 C | 3.35 B | 5.42 A | | 0.00D | 1.34 C | 5.32 B | 6.87 A | |
| | The second season (2019) | | | | | | | | | |
| IMZ (Control) | 0.00 e | 3.17 d | 6.85 b | 10.16 a | 5.04 A | 0.00 d | 4.44 c | 10.03 a | 10.23 a | 6.18 A |
| Wax | 0.00 e | 0.00 e | 3.44 cd | 6.63 b | 2.52 B | 0.00 d | 0.00 d | 6.64 b | 6.97 b | 3.40 B |
| not wax + 50 ppm nanosilver | 0.00 e | 0.00 e | 3.57 c | 3.47 cd | 1.76 C | 0.00 d | 0.00 d | 6.49 b | 6.42 b | 3.23 B |
| not wax + 100 ppm nanosilver | 0.00 e | 0.00 e | 3.27 d | 3.46 cd | 1.68 C | 0.00 d | 0.00 d | 3.84 c | 4.43 c | 2.07 C |
| wax + 100 ppm nanosilver | 0.00 e | 0.00 e | 0.00 e | 3.64 c | 0.91 D | 0.00 d | 0.00 d | 0.00 d | 6.67 b | 1.67 C |
| Mean | 0.00 D | 0.63 C | 3.43 B | 5.47 A | | 0.00 D | 0.89 C | 5.40 B | 6.94 A | |

IMZ = imazalil, lowercase letters in the same column indicate significant difference, while uppercase letters in the rows and columns indicate significant difference between means by LSD at a 0.05 level.

This data indicated that, compared with other treatments, coating with wax plus 100 ppm nanosilver and packaging in perforated polyethylene (PPE) was the best treatment for reducing the percentage of discarded fruit during the two seasons of a study in 2018 (Table 1) and 2019 (Table 1). Moreover, coating wax and nano silver at 50 or 100 ppm reduced discarded fruit percentage relative to the control, which gave the highest ratio. Coating with nano silver at 50 or 100 ppm significantly affected the discarded fruit percentage.

The control treatment could be seen, regardless of shelf-life period, to have the highest discarded fruit percentage compared to the combination of wax mixed with 100 ppm nanosilver and packaged in PPE, which recorded the lowest percentage. Moreover, all other treatments were more effective in reducing the discarded fruit percentage relative to

the control. Coating with wax and 50 ppm nanosilver and packaging in PPE treatments had a similar effect on discarded fruit percentage (Table 1).

Concerning the influence of the cold storage period on the discarded fruit percentage of Murcott mandarins, it was evident that after the second month of cold storage, the rate of discarded fruit increased as the cold storage period progressed, reaching the highest percentage after four months of cold storage (Table 1). The previous trend was typically repeated as the shelf-life period progressed in the two seasons of study.

The interaction between postharvest treatments and cold storage also affected the discarded fruit percentage. There was a significant increase in the discarded fruit percentage in the control treatment compared with other used treatments after the fourth month of cold Storage. On the other hand, coating treatments with nano silver—at both unmixed concentrations or at 100 ppm mixed with wax—were more effective in reducing the percentage of discarded fruit (similar in their effect) compared to using a wax coating alone or the control by the end of cold storage. This trend of results was nearly similar to that obtained with the interaction effect between postharvest treatments and shelf-life period, with one exception being that coating with nanosilver (100 ppm) alone before packaging in PPE had the greatest ability to reduce the percentage of discarded fruit relative to other treatments after the last period of shelf life in both seasons (Table 1).

*3.2. Weight Loss %*

Fruit weight loss is primarily connected to water loss, mainly because transpiration, which is responsible for 90% of overall weight reduction, initially originates from the peel [70–72]. The data in Table 2 demonstrate that during the 2018 and 2019 seasons, the percentage of weight lost generally rose with longer storage times in both the cold storage and shelf-life periods.

**Table 2.** Effect of postharvest treatments on fresh weight loss percentage (FWL%) of Murcott mandarin fruits during 1, 2, 3, and 4 months of cold storage and after 6 days of shelf life during the 2018 and 2019 seasons.

| Treatments | Cold Storage Period (Month) (P) | | | | | 6 Days Shelf Life (P) | | | | |
|---|---|---|---|---|---|---|---|---|---|---|
| | The First Season (2018) | | | | | | | | | |
| | 1 | 2 | 3 | 4 | Mean | 1 | 2 | 3 | 4 | Mean |
| IMZ (Control) | 1.80 e | 2.17 cd | 2.40 b | 2.90 a | 2.32 A | 2.03 ef | 2.30 cd | 2.60 b | 3.10 a | 2.51 A |
| Wax | 0.72 g | 1.17 f | 1.80 e | 2.23 bc | 1.48 C | 1.15 hi | 1.63 g | 2.00 f | 2.50 bc | 1.82 B |
| not wax + 50 ppm nanosilver | 0.87 g | 1.33 f | 2.13 cd | 2.23 bc | 1.64 B | 1.13 i | 1.50 g | 2.27 cde | 2.50 bc | 1.85 B |
| not wax + 100 ppm nanosilver | 0.77 g | 1.27 f | 2.00 d | 2.03 d | 1.52 C | 1.09 i | 1.43 g | 2.10 def | 2.57 b | 1.80 BC |
| wax + 100 ppm nanosilver | 0.75 g | 1.30 f | 1.70 e | 2.03 d | 1.44 C | 1.03 i | 1.40 gh | 1.97 f | 2.37 bc | 1.69 C |
| Mean | 0.98 D | 1.45 C | 2.01 B | 2.29 A | | 1.29 D | 1.65 C | 2.19 B | 2.61 A | |
| | The second season (2019) | | | | | | | | | |
| IMZ (Control) | 1.73 g | 2.33 bc | 2.50 b | 2.90 a | 2.37 A | 2.07 de | 2.53 b | 2.23 cd | 3.03 a | 2.47 A |
| Wax | 0.80 i | 1.30 h | 1.77 g | 2.30 cd | 1.54 C | 1.078 g | 1.63 f | 2.03 e | 2.47 b | 1.80 B |
| not wax + 50 ppm nanosilver | 0.87 i | 1.20 h | 2.23 cde | 2.27 cde | 1.64 B | 1.13 g | 1.47 f | 2.23 cd | 2.53 b | 1.84 B |
| not wax + 100 ppm nanosilver | 0.82 i | 1.27 h | 2.03 f | 2.13 def | 1.56 BC | 1.13 g | 1.47 f | 2.10 cde | 2.53 b | 1.81 B |
| wax + 100 ppm nanosilver | 0.91 i | 1.33 h | 1.73 g | 2.10 ef | 1.52 C | 1.07 g | 1.47 f | 2.03 e | 2.27 c | 1.71 C |
| Mean | 0.00 D | 0.63 C | 3.43 B | 5.47 A | | 0.00 D | 0.89 C | 5.40 B | 6.94 A | |

IMZ = imazalil, lowercase letters in the same column indicate significant difference, while uppercase letters in the rows and columns indicate significant difference between means by LSD at a 0.05 level.

Furthermore, the results showed that all applications significantly decreased weight loss compared to the control during both the cold storage period (Table 2) and days of shelf life (Table 2), and the applied treatments nanosilver (100 ppm) and wax with nanosilver

(100 ppm) with packaging in perforated polyethylene (PPE) were more effective.at reducing weight loss.

Furthermore, the most pronounced effect in reducing the weight loss percentage was recorded by combining a wax coating and 100 ppm nanosilver, and packaging the sample in PPE. Moreover, coating with wax and nanosilver at either 50 or 100 ppm reduced the fresh weight loss percentage of mandarins relative to the control.

In this respect, the data in Table 2 indicates the effect of the postharvest treatments, cold storage period, and their interaction with the fresh weight loss percentage of Murcott mandarins. This data showed that after the first month of cold storage (Table 2), all applied treatments were capable of reducing weight loss compared with the control. The differences among these treatments were not big enough to be significant except for "coated with 100 ppm nanosilver"—alone or mixed with wax and packaged in PPE. The weight loss percentage for all applied treatments tended to increase significantly with the advancement of cold storage.

Similar results were nearly found when discussing the interaction effect between used postharvest treatments and shelf-life (Table 2) duration, except for the combination consisting of wax plus 100 ppm nanosilver and packaging in PPE, which was able to record the lowest weight loss percentage as compared with other treatments after the last period of shelf life, especially in the second season (Table 2).

### 3.3. Pulp Firmness

The strength and fruit hardness of coated mandarins were significantly improved. In comparison, uncoated fruit undergoes tissue suppleness with time while being stored [53,73]. For both coated and uncoated fruits, there is no discernible change in fruit firmness during the first few days of low-temperature storage; rather, variations emerge over time [74].

The effect of postharvest treatments during the cold storage period on pulp firmness during the 2018 and 2019 seasons is displayed in Table 3. The data revealed that the combination consisting of coating with wax plus 100 ppm nanosilver and packaging in perforated polyethylene (PPE) was the most effective treatment for reducing the loss of pulp firmness, and it gave the greatest value of firmness as compared with other used treatments. In addition, all applied treatments recorded a higher firmness value than the control treatment, which showed the lowest value. This trend was stable in the two seasons of study (Table 3).

During the shelf-life period, it could be noticed that coating with wax mixed with 100 ppm nanosilver and packaging in PPE had a more elevated value for pulp firmness. Moreover, other treatments also caused a higher firmness value relative to the control treatment in both seasons (Table 3). On the contrary, mandarins treated with imazalil and packaged in PPE had the lowest firmness value.

### 3.4. Vitamin C Content

Due to acid consumption as respiration substrates, ascorbic acid degrades over time when stored [75]. Nevertheless, the data shown in Table 4 indicated that the highest content of vitamin C in the juice of Murcott mandarin was obtained by coating with either wax, 50 ppm nanosilver, or the combination of wax and 100 ppm nanosilver treatments with packaging in perforated polyethylene (PPE) during cold storage period effect in the two seasons of study (Table 4). The highest impact was demonstrated by the last treatment (the combination of wax and 100 ppm nanosilver), which maintained the highest content of vitamin C. In contrast, both seasons found the least vitamin C content in the control treatment (imazalil followed by packaging in PPE).

The same trend was noticed throughout the shelf-life periods (Table 4) consistently during the two seasons of study. All used treatments recorded higher vitamin C content than the control. Moreover, samples coated in all treatments with wax and others before packaging in PPE were similar in their vitamin C content in the 2019 season (Table 4).

**Table 3.** Effect of some postharvest applied treatments on pulp firmness (g/cm$^2$) of Murcott mandarin fruits during 1, 2, 3, and 4 months of cold storage and after 6 days of shelf life during 2018 and 2019 seasons.

| Treatments | Cold Storage Period (Month) (P) | | | | | 6 Days Shelf Life (P) | | | | |
| --- | --- | --- | --- | --- | --- | --- | --- | --- | --- | --- |
| | The First Season (2018) | | | | | | | | | |
| | 1 | 2 | 3 | 4 | Mean | 1 | 2 | 3 | 4 | Mean |
| IMZ (Control) | 180.00 b | 165.00 c | 141.67 e | 115.00 g | 150.42 C | 165.00 bc | 145.00 ef | 137.33 f | 106.67 i | 138.50 C |
| Wax | 190.00 a | 173.00 b | 145.00 de | 125.00 f | 158.25 B | 170.00 ab | 160.33 bc | 144.33 ef | 113.33 hi | 148.75 AB |
| not wax + 50 ppm nanosilver | 193.33 a | 175.00 b | 144.67 de | 120.00 fg | 158.25 B | 170.00 ab | 165.00 bc | 142.33 ef | 110.00 i | 146.83 B |
| not wax + 100 ppm nanosilver | 190.00 a | 173.67 b | 150.00 d | 127.00 f | 160.17 B | 165.00 bc | 155.00 d | 145.00 ef | 120.00 gh | 146.25 B |
| wax + 100 ppm nanosilver | 190.00 a | 175.00 b | 163.33 c | 148.33 de | 169.17 A | 175.33 a | 160.00 cd | 145.67 e | 123.33 g | 151.08 A |
| Mean | 188.67 A | 172.33 B | 148.93 C | 127.07 D | | 169.07 A | 158.47 B | 142.93 C | 114.67 D | |
| | The second season (2019) | | | | | | | | | |
| IMZ (Control) | 180.00 b | 160.00 e | 141.00 g | 113.33 i | 148.58 C | 160.00 cd | 141.67 ef | 137.67 f | 103.33 h | 135.67 C |
| Wax | 195.00 a | 173.00 cd | 147.67 f | 128.33 h | 161.00 B | 175.00 a | 165.67 bc | 145.33 e | 115.00 g | 150.25 A |
| not wax + 50 ppm nanosilver | 192.33 a | 171.67 d | 146.67 f | 123.33 h | 158.50 B | 172.67 ab | 156.67 d | 141.33 ef | 113.33 g | 146.00 B |
| not wax + 100 ppm nanosilver | 191.00 a | 167.67 d | 151.67 f | 126.67 h | 159.25 B | 169.33 ab | 158.33 d | 148.33 e | 116.67 g | 148.17 AB |
| wax + 100 ppm nanosilver | 190.67 a | 177.33 bc | 167.00 d | 148.00 f | 170.75 A | 172.33 ab | 161.00 cd | 147.67 e | 118.33 g | 149.83 A |
| Mean | 189.80 A | 169.93 B | 150.80 C | 127.93 D | | 169.87 A | 156.67 A | 144.07 A | 113.33 B | |

IMZ = imazalil, lowercase letters in the same column indicate significant difference, while uppercase letters in the rows and columns indicate significant difference between means by LSD at a 0.05 level.

**Table 4.** Effect of some postharvest applied treatments on vitamin C content (mg/100 mL juice) of Murcott mandarin fruits during 1, 2, 3, and 4 months of cold storage and after 6 days of shelf life during 2018 and 2019 seasons.

| Treatments | Cold Storage Period (Month) (P) | | | | | 6 Days Shelf Life (P) | | | | |
| --- | --- | --- | --- | --- | --- | --- | --- | --- | --- | --- |
| | The First Season (2018) | | | | | | | | | |
| | 1 | 2 | 3 | 4 | Mean | 1 | 2 | 3 | 4 | Mean |
| IMZ (Control) | 51.27 bc | 50.00 d | 47.53 f | 43.17 h | 47.99 C | 50 bcd | 47.76 gh | 44.93 kl | 41.50 n | 46.05 C |
| Wax | 51.63 b | 50.67 bcd | 47.93 ef | 45.00 g | 48.81 AB | 50.60 ab | 48.90 ef | 45.77 jk | 42.00 n | 46.82 B |
| not wax + 50 ppm nanosilver | 52.90 a | 50.50 bcd | 48.23 ef | 45.20 g | 49.21 A | 50.40 abc | 48.17 fg | 46.03 ij | 43.67 m | 47.07 B |
| not wax + 100 ppm nanosilver | 50.43 cd | 50.33 cd | 48.63 e | 43.73 h | 48.28 BC | 50.03 bcd | 49.20 de | 45.13 jkl | 41.93 n | 46.57 B |
| wax + 100 ppm nanosilver | 51.67 b | 50.10 cd | 47.57 f | 45.57 g | 48.72 AB | 51.03 a | 49.60 cde | 47.00 hi | 44.23 lm | 47.97 A |
| Mean | 51.58 A | 50.32 B | 47.98 C | 44.53 D | | 50.41 A | 48.73 B | 45.77 C | 42.67 D | |
| | The second season (2019) | | | | | | | | | |
| IMZ (Control) | 50.00 de | 49.10 ef | 47.70 h | 42.00 k | 47.20 B | 49.50 bc | 47.87 ef | 45.10 ij | 40.83 m | 45.82 D |
| Wax | 51.33 bc | 50.33 cd | 48.23 fgh | 44.90 ij | 48.70 A | 50.10 ab | 49.33 bcd | 45.90 hi | 42.23 l | 46.89 BC |
| not wax + 50 ppm nanosilver | 53.00 a | 50.27 cd | 48.40 fgh | 44.70 ij | 49.09 A | 50.67 a | 48.63 de | 46.27 gh | 43.27 k | 47.21 B |
| not wax + 100 ppm nanosilver | 51.06 bcd | 50.33 cd | 48.83 efg | 43.90 j | 48.53 A | 50.00 abc | 49.23 cd | 45.33 i | 41.40 lm | 46.49 C |
| wax + 100 ppm nanosilver | 52.00 ab | 50.23 cd | 47.83 gh | 45.63 i | 48.92 A | 50.00 abc | 49.17 cd | 47.07 fg | 44.47 j | 47.67 A |
| Mean | 51.48 A | 50.05 B | 48.20 C | 44.23 D | | 50.05 A | 48.84 B | 45.93 C | 42.44 D | |

IMZ = imazalil, lowercase letters in the same column indicate significant difference, while uppercase letters in the rows and columns indicate significant difference between means by LSD at a 0.05 level.

### 3.5. Panel Taste Index

In recent studies, 46 different mandarin varieties belonging to several natural subgroups were examined for wide genetic variability in numerous fruit-quality features, including physical, physiological (ripening period), nutritional composition, and sensory attributes [76].

Nevertheless, the effects of some postharvest treatments in the cold storage period are shown in Table 5. The data indicated that all used treatments gave an excellent taste index for Murcott mandarins, with significant differences relative to the control treatment in both seasons. The same trend was observed with the effect of used postharvest treatments throughout the shelf-life period in both study seasons (Table 5).

**Table 5.** Effect of some postharvest applied treatments on taste panel index (PTI) of Murcott mandarin fruits during 1, 2, 3, and 4 months of cold storage and after 6 days of shelf life during 2018 and 2019 seasons.

| Treatments | Cold Storage Period (Month) (P) | | | | | 6 Days Shelf Life (P) | | | | |
|---|---|---|---|---|---|---|---|---|---|---|
| | The First Season (2018) | | | | | | | | | |
| | 1 | 2 | 3 | 4 | Mean | 1 | 2 | 3 | 4 | Mean |
| IMZ (Control) | 4.33 abc | 3.00 ef | 2.33 f | 1.33 g | 2.75 B | 3.67 cd | 3.00 ef | 2.00 g | 1.00 h | 2.42 B |
| Wax | 4.67 ab | 4.00 bcd | 3.33 de | 3.00 ef | 3.75 A | 4.00 bc | 3.33 de | 3.00 ef | 2.67 f | 3.25 A |
| not wax + 50 ppm nanosilver | 5.00 a | 4.00 bcd | 4.00 bcd | 3.00 ef | 4.00 A | 4.33 ab | 3.67 cd | 3.00 ef | 3.00ef | 3.50 A |
| not wax + 100 ppm nanosilver | 5.00 a | 4.00 bcd | 4.00 bcd | 3.33 de | 4.08 A | 4.67 a | 3.00 ef | 3.33 de | 2.00 g | 3.25 A |
| wax + 100 ppm nanosilver | 5.00 a | 4.00 bcd | 3.67 cde | 3.33 de | 4.00 A | 4.00 bc | 3.67 cd | 3.33 de | 3.00 ef | 3.50 A |
| Mean | 4.80 A | 3.80 B | 3.47 B | 2.80 C | | 4.13 A | 3.33 B | 2.93 C | 2.33 D | |
| | The second season (2019) | | | | | | | | | |
| IMZ (Control) | 4.00 bcd | 3.33 def | 2.33 g | 1.33 h | 2.75 B | 4.00 abc | 3.00 de | 2.00 f | 1.00 g | 2.50 B |
| Wax | 4.67 ab | 4.00 bcd | 3.67 cde | 3.00 efg | 3.83 A | 4.33 ab | 3.67 bcd | 3.00 de | 2.67 ef | 3.42 A |
| not wax + 50 ppm nanosilver | 5.00 a | 4.33 abc | 3.67 cde | 2.67 fg | 3.92 A | 4.67 a | 3.33 cde | 3.00 de | 2.67 ef | 3.42 A |
| not wax + 100 ppm nanosilver | 5.00 a | 4.33 abc | 4.00 bcd | 3.33 def | 4.17 A | 4.67 a | 3.67 bcd | 3.67 bcd | 2.67 ef | 3.67 A |
| wax + 100 ppm nanosilver | 5.00 a | 4.33 abc | 3.67 cde | 3.00 efg | 4.00 A | 4.00 abc | 3.33 cde | 3.33 cde | 3.00 de | 3.42 A |
| Mean | 4.73 A | 4.07 B | 3.47 C | 2.67 C | | 4.33 A | 3.40 B | 3.00 C | 2.40 D | |

IMZ = imazalil, lowercase letters in the same column indicate significant difference, while uppercase letters in the rows and columns indicate significant difference between means by LSD at a 0.05 level.

Concerning the influence of the cold storage period and shelf-life period on the taste panel index, the results introduced in Table 5 showed that the "excellent" taste panel index was noted after the first month of cold storage. Subsequently, the taste panel index decreased gradually by the end of cold storage in both seasons.

### 3.6. Catalase Enzyme Activity

The response of catalase enzyme activity in Murcott mandarin to various postharvest treatments, regardless of the cold storage period, is displayed in Figure 1A. The results showed that the highest catalase enzyme activity was found in the control (imazalil) and wax, compared with other remaining treatments. The combined wax—100 ppm nanosilver showed the lowest activity of catalase enzymes.

Concerning the influence of the cold storage period on catalase enzyme activity, the data shown in Figure 1A indicated that catalase enzyme activity after two months of cold storage was higher than after four months of that storage.

Catalase enzyme activity is also affected by the interaction between postharvest treatments and the cold storage period, where the treatments of coating with either 100 or 50 ppm nanosilver gave high catalase enzyme activity after two months of cold storage. On the other hand, the lowest activity of the catalase enzyme was recorded with the samples coated with wax plus 100 ppm nano silver after four months of cold storage.

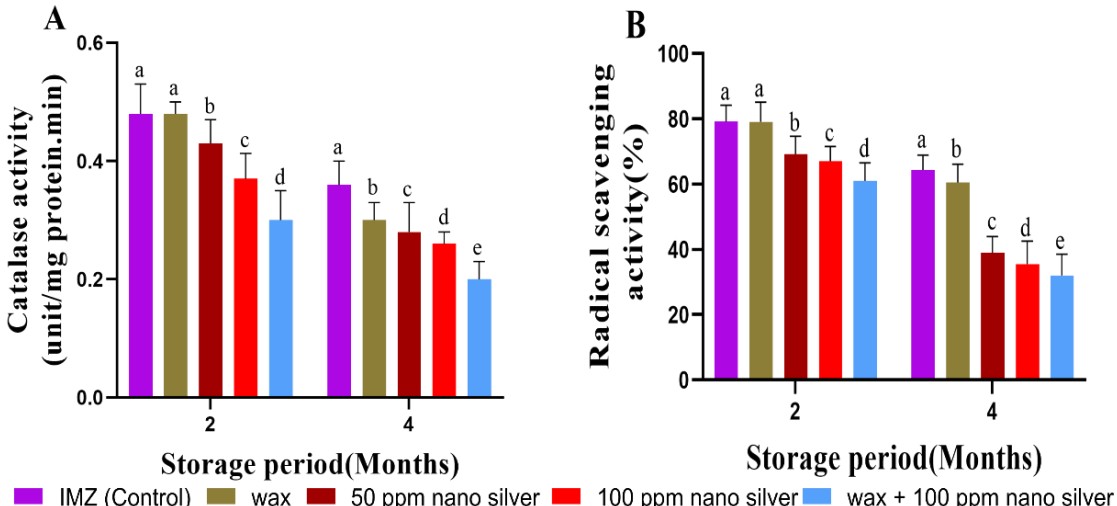

**Figure 1.** Studying the effect of some applied treatments after harvesting on catalase enzyme activity (**A**) and total antioxidant (**B**) of Murcott mandarin fruits during 2, and 4 months of cold storage during 2019 season during cold storage periods during the 2019 season. Data presented as mean $\pm$ SD, different lowercase letters above columns indicate significant differences.

### 3.7. Total Antioxidant Activities

The data in Figure 1B also indicated postharvest treatments' effect on total oxidants. The treatments coating with wax, imazalil, and the combination of wax plus 100 ppm nano silver, respectively showed that the total oxidants of Murcott mandarins after two months of cold storage were higher than after four months.

After two months of cold storage, Murcott mandarin fruits coated with wax and imazalil had higher total antioxidants than those coated with other treatments. Conversely, coating with 50 and 100 ppm nanosilver and combined wax–100 ppm nanosilver treatments gave the lowest total antioxidants. However, the combined wax—100 ppm nanosilver had the lowest total antioxidants relative to other used treatments.

### 3.8. Total Acidity Percentage

During the cold storage period, data were clear that the total acidity percentage (g citric acid/100 mL juice) gradually and significantly decreased with the advance in the cold storage period in the two seasons (Table 6). The lowest values were recorded four months after cold storage in the two seasons, while the highest values resulted from treatments after one month of cold storage. Coating with wax and 100 ppm nanosilver treatment retained a significantly higher acidity percentage compared with the control and other treatments in both seasons. The interaction between cold storage and the studied treatments was significant in the two seasons. The highest value in the first and second seasons (1.097 and 1.1%, respectively) came from coating with wax and 100 ppm nanosilver at one month. The lowest value in the two seasons (0.495 and 0.493% respectively) came from the control at the end of cold storage.

After the shelf-life period, the data also show that the total acidity percentage was markedly decreased with the advance in the shelf-life period (Table 6). In the two seasons, the lowest percentage was recorded during shelf life after four months of cold storage. All treatments retained significantly higher total acidity compared with the control in both seasons. Additionally, there were significant differences between all the coating with nano silver and the coating with wax-only treatments in the two seasons. The interaction between treatments and the shelf-life period was significant in both seasons. The lowest percentage during shelf life always came from treatments tested after four months of cold storage.

**Table 6.** Effect of Nanosilver coating on total acidity percentage (TA %) of Murcott mandarin fruits during 1, 2, 3, and 4 months of cold storage and after 6 days of shelf life during 2018 and 2019 seasons.

| Treatments | Cold Storage Period (Month) (P) | | | | | 6 Days Shelf Life (P) | | | | |
|---|---|---|---|---|---|---|---|---|---|---|
| | The First Season (2018) | | | | | | | | | |
| | **1** | **2** | **3** | **4** | **Mean** | **1** | **2** | **3** | **4** | **Mean** |
| IMZ (Control) | 0.900 c | 0.650 fgh | 0.600 ghi | 0.495 j | 0.661 D | 0.750 d | 0.656 fg | 0.507 jk | 0.467 l | 0.595 E |
| Wax | 0.947 bc | 0.733 de | 0.650 fgh | 0.573 i | 0.726 C | 0.900 c | 0.687 e | 0.565 hi | 0.503 k | 0.664 D |
| not wax + 50 ppm nanosilver | 0.980 b | 0.757 d | 0.670 ef | 0.550 ij | 0.739 C | 0.947 b | 0.666 efg | 0.570 h | 0.533 ij | 0.679 C |
| not wax + 100 ppm nanosilver | 1.100 a | 0.750 d | 0.650 fgh | 0.593 hi | 0.773 B | 0.980 a | 0.656 fg | 0.643 g | 0.580 h | 0.715 B |
| wax + 100 ppm nanosilver | 1.097 a | 0.933 bc | 0.750 d | 0.660 fg | 0.860 A | 0.953 ab | 0.770 d | 0.686 ef | 0.553 hi | 0.741 A |
| Mean | 1.005 A | 0.765 B | 0.664 C | 0.574 D | | 0.906 A | 0.687 B | 0.594 C | 0.527 D | |
| New LSD at 0.05% | | T = 0.031 | | P = 0.066 T × P = 0.063 | | | T = 0.014 P = 0.023 T × P = 0.028 | | | |
| | The second season (2019) | | | | | | | | | |
| IMZ (Control) | 0.980 c | 0.633 gh | 0.603 hi | 0.493 j | 0.677 D | 0.750 c | 0.603 ef | 0.503 ij | 0.477 j | 0.583 E |
| Wax | 1.030 b | 0.710 f | 0.680 fg | 0.573 hi | 0.766 BC | 0.907 b | 0.667 d | 0.570 fg | 0.500 ij | 0.661 D |
| not wax + 50 ppm nanosilver | 0.992 bc | 0.783 e | 0.687 fg | 0.543 i | 0.751 C | 0.960 a | 0.660 d | 0.570 fg | 0.523 hi | 0.678 C |
| not wax + 100 ppm nanosilver | 1.033 b | 0.797 e | 0.693 fg | 0.597 hi | 0.780 B | 0.982 a | 0.667 d | 0.617 e | 0.557 gh | 0.705 B |
| wax + 100 ppm nanosilver | 1.100 a | 0.910 d | 0.773 e | 0.613 h | 0.832 A | 0.960 a | 0.777 c | 0.690 d | 0.563 g | 0.747 A |
| Mean | 1.027 A | 0.767 B | 0.687 C | 0.564 D | | 0.912 A | 0.675 B | 0.590 C | 0.524 D | |
| New LSD at 0.05% | T = 0.022 P = 0.046 T × P = 0.044 | | | | | | T = 0.017 P = 0.013 T × P = 0.034 | | | |

T = Treatment, P = period, T × P= interaction between treatments and period, IMZ = imazalil, lowercase letters in the same column indicate significant difference, while uppercase letters in the rows and columns indicate significant difference between means by LSD at a 0.05 level.

### 3.9. Total Soluble Solids (TSSs)

During the cold storage period, it is clear that TSSs increased with the advance of the cold storage period in both seasons (Table 7). Moreover, TSSs were significantly affected by the tested treatments. The wax coated–100 ppm nanosilver treatment recorded the fewest TSSs in both seasons compared with other treatments. The interaction between the studied treatments and the cold storage period was significant in the two seasons.

After the shelf-life period, the total soluble solids increased during shelf-life as the cold storage period advanced (Table 7). The control treatment recorded the highest TSS in the two seasons compared with other treatments. The wax coated–100 ppm nanosilver treatment recorded the fewest TSSs in both seasons compared with other treatments.

**Table 7.** Effect of Nanosilver coating on total soluble solids (TSS) (Brix°) of Murcott mandarin fruits during 1, 2, 3, and 4 months of cold storage and after 6 days of shelf life during the 2018 and 2019 seasons.

| Treatments | Cold Storage Period (Month) (P) | | | | | 6 Days Shelf Life (P) | | | | |
|---|---|---|---|---|---|---|---|---|---|---|
| | The First Season (2018) | | | | | | | | | |
| | 1 | 2 | 3 | 4 | Mean | 1 | 2 | 3 | 4 | Mean |
| IMZ (Control) | 9.50 gh | 10.07 de | 10.50 ab | 10.70 a | 10.19 A | 9.80 fg | 10.33 cd | 10.40 bcd | 10.90 a | 10.36 A |
| Wax | 9.07 i | 9.50 gh | 10.30 abc | 10.50 ab | 9.84 BC | 9.53 g | 10.00 ef | 10.53 bcd | 10.63 abc | 10.17 B |
| not wax + 50 ppm nanosilver | 8.83 jk | 9.80 e–h | 10.00 def | 10.20 bcd | 9.71 CD | 9.03 h | 10.00 ef | 10.47 bcd | 10.87 a | 10.09 B |
| not wax + 100 ppm nanosilver | 9.40 h | 9.90 d–g | 10.10 cde | 10.50 ab | 9.97 B | 9.50 g | 10.00 ef | 10.30 de | 10.70 ab | 10.12 B |
| wax + 100 ppm nanosilver | 8.50 k | 9.60 fgh | 10.00 def | 10.20 bcd | 9.57 D | 9.00 h | 9.80 fg | 10.27 de | 10.53 bcd | 9.90 C |
| Mean | 9.06 D | 9.77 C | 10.18 B | 10.42 A | | 9.37 D | 10.03 C | 10.39 B | 10.73 A | |
| New LSD at 0.05% | T = 0.19 P = 0.21 T × P = 0.39 | | | | | T = 0.14 P = 0.16 T × P = 0.29 | | | | |
| | The second season (2019) | | | | | | | | | |
| IMZ (Control) | 9.70 fgh | 10.03 cde | 10.47 ab | 10.77 a | 10.24 A | 9.87 e | 10.87 a | 10.30 d | 10.63 b | 10.42 A |
| Wax | 9.50 hi | 9.60 ghi | 10.23 bcd | 10.47 ab | 9.95 B | 9.67 f | 10.00 e | 10.47 bcd | 10.67 ab | 10.20 B |
| not wax + 50 ppm nanosilver | 8.97 j | 9.83 efg | 10.07 cde | 10.33 bc | 9.80 B | 9.03 g | 10.03 e | 10.53 bc | 10.83 a | 10.11 B |
| not wax + 100 ppm nanosilver | 9.30 i | 9.83 efg | 10.00 def | 10.47 ab | 9.90 B | 9.50 f | 10.03 e | 10.37 cd | 10.63 b | 10.13 B |
| wax + 100 ppm nanosilver | 8.50 k | 9.67 gh | 10.07 cde | 10.23 bcd | 9.62 C | 9.13 g | 9.97 e | 10.33 d | 10.56 bc | 10.00 C |
| Mean | 9.19 D | 9.79 C | 10.17 B | 10.45 A | | 9.44 D | 10.18 C | 10.40 B | 10.67 A | |
| New LSD at 0.05% | T = 0.15 P = 0.17 T × P = 0.31 | | | | | T = 0.09 P = 0.14 T × P = 0.19 | | | | |

T = Treatment, P = period, T × P = interaction between treatments and period, IMZ = imazalil, lowercase letters in the same column indicate significant difference, while uppercase letters in the rows and columns indicate significant difference between means by LSD at a 0.05 level.

## 4. Discussion

One of the most important factors that negatively affect fruit quality is water loss, which reduces its commercial life after harvest [77]. For orange fruits, a 2.5% weight loss causes a contraction to begin, and a 5% loss of its original weight makes it no longer marketable [78]. Murcott mandarin fruit weight loss increased progressively in all treatments with the increasing storage period, as shown in Table 2. When separated from the tree, mature fruits undergo a number of metabolic processes such as transpiration and respiration, and there is a positive relationship between weight loss and the rate of respiration and transpiration [79]. The activity of metabolic processes in fruits leads to weight loss and fruit quality during the storage period and shelf life [12,80]. Coatings can reduce water loss and thus reduce harmful effects by trapping the moisture inside the fruits. In addition to preventing the exit of water vapor from the stomata on the peel (reducing the transpiration process) and thus maintaining the firmness of the fruit, there are many studies showing that the use of nanoparticles causes fresh weight preservation [81,82]. As a result, keeping fruit in cold and humid environments has a major impact on stomatal behavior and lowers the rate of water loss during storage. High temperatures result in peel shriveling, drying, a dull look, softening, and peel senescence in the end [83,84].

Citrus fruit firmness reveals the thickness and turgidity of the peel [85]. Fruit pulp firmness, which is a crucial factor for the quality of the fruits postharvest, was noticeably decreased during the storage period in all treatments (Table 3). In ripe fruit, reduced fruit firmness with maturing is often associated with the breakdown of the pectic components

of the cell wall. Mostly, this is not the first reason for the softening of citrus fruits, in which the dissolution of pectin with ripening is very slow [86]. The reduced firmness of citrus fruits is mainly related to the loss of water from the peel, development, and senescence [87], as well as pathogens that infect the peel and secrete the enzyme that degrades the cell wall [88]. The reason the hardness of the pulp in the coated fruits was maintained is due to a decrease in the process of transpiration and respiration and a delay in rapid ripening during storage. The nano-coating material also effectively contributes to inhibiting the enzymatic and metabolic activities in the fruit and resisting the fungal infections that affect citrus [77,79,88].

Ascorbic acid (vitamin C) concentration in fruits decreases with prolonged storage, as organic acids are consumed as substrates in respiration [75,89]. Despite this, the nano-coating material plus wax (wax + 100 ppm nanosilver) was better, as it kept the level of ascorbic acid above the control level throughout storage in both seasons (Table 4). Increased water loss in fruits leads to rapid oxidation, and, therefore, a rapid loss of ascorbic acid [90]. In other studies, it was found that using high concentrations of nanomaterials in coating formulations, significantly maintained the level of ascorbic acid in coated fruits [91]. The organic acids in fruits decrease during postharvest storage as a result of their use as metabolic substrates in the respiratory system [92,93]. The combined wax–100 ppm nanosilver coating treatment retained a significantly higher acidity percentage compared with the control and other treatments in the two seasons. This may be because the coating inhibited the activity of metabolic enzymes and slowed down the rate of acidolysis in pears during storage [94].

TSSs were significantly affected by the tested treatments. The coated with wax and 100 ppm nanosilver treatment recorded the fewest TSSs in both seasons compared with other treatments. This decrease in soluble solids in the covered fruits is attributed to the slower metabolic processes, such as respiration and transpiration, compared to the untreated fruits of various postharvest treatments [95].

Previous studies showed that the taste panel of mandarin varieties coated with a low gas permeability layer has a less fresh flavor compared to those covered with a higher gas permeability layer (polyethylene and wax) [96,97]. The panel taste index found that the nanoparticle-coated fruit had more tangerine flavor than the uncoated [96]. These results were partially in agreement with those obtained by [43].

The obtained results may be due to the wax coatings contributing to the fruit shine as well as maintaining gaseous exchange and water retention. The fruit continues to respire after harvest, and although the content and composition of coatings provide high levels of wax gloss, they tend to negatively affect the permeation of gases through the peel, which might lead to the development of off-flavors [15,52,98].

The typical increased off flavor volatiles associated with anaerobic respiration in the fruit include ethanol and acetaldehyde [52,99]. Furthermore, the wax application plays an important role in prolonging fruit quality, with differing effects on some fruit quality parameters [26]. Thus, the activity of catalase enzyme and total antioxidant activities decreases. Moreover, nano silver particles, considered an antibacterial agent, promise longer durability for food [100], and nano silver particles inhibited mycelium growth of *Penicillium digitatum* and *Aspergillus niger* during storage [43], and nano silver particle formulated mucilage exhibited bactericidal activity for *Escherichia coli* and *Staphylococcus* as well as inhibited growth of *Fusarium solani* and *Aspergillus niger* [101]. Nanosilver particles significantly controlled microbial proliferation and could be considered a biocidal preservative [102]. Furthermore, as ethylene signaling inhibitors, nano silver particles effectively reduce ethylene content to increase life commercially [103].

## 5. Summary

The short shelf life of citrus fruits during storage has a significant impact on the determinants of fruit quality. Recently, the use of a variety of harmless and usable coatings, such as plant extracts as well as nanomaterials and others, to extend the shelf life of fruits and vegetables has been widely used. In this study, we examined several different combinations of wax and nanosilver to coat Murcott mandarin fruits during storage and shelf life, and we examined the overall effect of these coatings on quality evaluation during 1, 2, 3, and 4 months of cold storage and after 6 days of shelf life during two seasons. From the obtained data, it could be proven that the combined wax—100 ppm nanosilver and packaged in 0.005% perforated polyethylene (PPE) treatment was the most effective treatment. Therefore, these coatings could be promising alternative materials for extending mandarin fruits' postharvest life and marketing period.

**Author Contributions:** M.M.G. (Mohamed M. Gemail), I.E.E. and M.M.G. (Mohamed M. Gad) performed the experiments with support from I.E.E., S.M.E.-H. and M.M.G. (.Mohamed M Gemail) conceived the project. S.M.E.-H. and M.M.G. (Mohamed M. Gad) designed the experiments. I.E.E., C.C., M.M.J., B.A., W.A.A. and M.M.G. (Mohamed M. Gemail) analyzed the data and wrote the manuscript. All authors have read and agreed to the published version of the manuscript.

**Funding:** This research received no external funding.

**Institutional Review Board Statement:** Not applicable.

**Informed Consent Statement:** Not applicable.

**Data Availability Statement:** Not applicable.

**Acknowledgments:** All authors are grateful to the Horticulture Department, Faculty of Agriculture, Zagazig University, Egypt for providing some facilities and equipment to perform this work.

**Conflicts of Interest:** The authors declare no conflict of interest.

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
