# Peer review of "Influence of Wax and Silver Nanoparticles on Preservation Quality of Murcott Mandarin Fruit during Cold Storage and after Shelf-Life"

_coatings, doi:10.3390/coatings13010090_

Round 1

Reviewer 1 Report

Briefly, the authors bring forward the proposal to examine the influence of wax and nanosilver treatments on the preservation of Murcott mandarin fruit during storage and after shelf-life. The results obtained could help fruit producers to reduce product loss after harvesting.

Nevertheless, the manuscript presents important issues that must be addressed before publication. 

I have provided more details and comments below:

1- English must be extensively revised throughout the manuscript.

2- In line 41, please replace the provided unit of area for one of the International System of Units (For instance, Km2).

3- Text formatting needs to be carefully revised (Formatting consistency, paragraph spacing, adequate use of capital letters, font size, etc). Some examples that must be addressed can be found in lines 86, 99, 161, 168, 187, and 354.

4- Please, maintain unit consistency throughout the manuscript. For example, the liter volume unit is described as both L and l in the text.

5- Please, describe in more detail the production of the nanosilver solutions used in the present study. For instance, what was the solvent used?

6- Could you further explain the mechanism of antioxidant activity loss on the studied fruits, considering the treatments applied? Why was the lowest antioxidant activity observed in the treatment combining wax plus 100 ppm nanosilver?

7- What do the authors mean by “trapping the soil inside the fruits” in line 146?

Author Response

Dear reviewer, Thank you for this complete review, which contributes greatly to the improvement of the manuscript.

  • English must be extensively revised throughout the manuscript.

Answer: - done; the linguistic review was done by a friend whose native language is English.

  • In line 41, please replace the provided unit of area for one of the International System of Units (For instance, Km2).

Answer: - done; the area unit has been replaced by km2, one of the International System units.

  • Text formatting needs to be carefully revised (Formatting consistency, paragraph spacing, adequate use of capital letters, font size, etc). Some examples that must be addressed can be found in lines 86, 99, 161, 168, 187, and 354.

Answer: - done.

  • Please, maintain unit consistency throughout the manuscript. For example, the liter volume unit is described as both L and l in the text.

Answer: - done; the liter volume unit is described as L throughout the manuscript.

  • Please, describe in more detail the production of the nanosilver solutions used in the present study. For instance, what was the solvent used?

Answer: - done; we used aqueous extracts to obtain the phenolic compounds from pomegranate and watermelon peels, which we used to convert silver nitrate to silver nanoparticles.

  • Could you further explain the mechanism of antioxidant activity loss on the studied fruits, considering the treatments applied? Why was the lowest antioxidant activity observed in the treatment combining wax plus 100 ppm nanosilver?

Answer: - The presence of wax and nanosilver in one mixture forms a coating for the fruit leads to maintaining significant stability in the quality characteristics of the fruit during cold storage as well as during shelf life, as it leads to a decrease in the activity of the catalase enzyme and thus a reduction in the activity of antioxidants, in addition to that, the silver nanoparticles prevent the growth of Fungi as well as many types of harmful bacteria that cause the deterioration of the quality of the fruit. Thus, it is possible to benefit from using silver nanoparticles with wax better than wax separately as a coating for the fruit.

  • What do the authors mean by “trapping the soil inside the fruits” in line 146?

Answer: - done; here is a fault the water has been replaced by soil; this sentence is correct (trapping the moisture inside the fruits).

Reviewer 2 Report

Dear Authors,

Detailed notes on the manuscript are given below:

1) Abstarct - provide the purpose of the work

2) How was the minimum fruit sample size calculated for the experiment?

3) Lines 161 and 163 - start a sentence with capital letters

4) There are unnecessary bold fonts in the text (applies to the entire manuscript, correct it)

5) What kind of statistical analysis was used?

6) In Tables 1-7 you provide letter designations (a, b, c ... etc. - I presume these are homogeneous group designations) which indicates that you used parametric tests (perhaps ANOVA). Parametric tests must be preceded by testing the normality of the population distribution of empirical variables and the homogeneity of variance in the samples - have such tests been performed, what was their result?

7) Figure 1. Effect of some postharvest … - font size, what do the vertical error bars mean (Sd+mean, 5%, Sd - describe it in the methodology)?

8) Chapter 5. Conclusions - is a summary, I suggest changing it to "Summary"

9) I also suggest that other methods of protecting the periderm of the crop (potato tubers, apples, etc.; microwaves, UV-C, magnetic field, etc.) should be mentioned in the introduction - some older literature items can be replaced with newer studies.

The manuscript needs clarification on the statistical analysis of the data.

Author Response

Dear reviewer, thank you for this complete review, which contributes greatly to the improvement of the manuscript.

  • Abstract - provide the purpose of the work

Answer: - done.

  • How was the minimum fruit sample size calculated for the experiment?

Answer: - we calculated sample size from equation 

  • Lines 161 and 163 - start a sentence with capital letters

Answer: - done.

  • There are unnecessary bold fonts in the text (applies to the entire manuscript, correct it)

Answer: - done.

  • What kind of statistical analysis was used?

Answer: -ANOVA test at a confidence level of 95%

  • In Tables 1-7 you provide letter designations (a, b, c ... etc. - I presume these are homogeneous group designations) which indicates that you used parametric tests (perhaps ANOVA). Parametric tests must be preceded by testing the normality of the population distribution of empirical variables and the homogeneity of variance in the samples - have such tests been performed, what was their result?

Answer: - The triplicate data means were analyzed for statistical differences by one-way ANOVA at confidence level of 95% [67], using Costat program version 6.4 (Costat 2008). The sample size was calculated from the following equation (3)

 Means were compared with the least significant difference (LSD) as a post hoc test at a probability level of 5%.

  • Figure 1. Effect of some postharvest … - font size, what do the vertical error bars mean (Sd+mean, 5%, Sd - describe it in the methodology)?

Answer: - Data presented as mean ±SD, different lowercase letters above columns indicate significant differences

  • Chapter 5. Conclusions - is a summary, I suggest changing it to "Summary"

Answer: - done.

  • I also suggest that other methods of protecting the periderm of the crop (potato tubers, apples, etc.; microwaves, UV-C, magnetic field, etc.) should be mentioned in the introduction - some older literature items can be replaced with newer studies.

Answer: - done.

  • The manuscript needs clarification on the statistical analysis of the data.

Answer: - done, the statistical analysis is improved in the manuscript.

Reviewer 3 Report

Thanks for good work. Comments are mentioned in the attached file.

Author Response

Dear reviewer, thank you for this complete review, which contributes greatly to the improvement of the manuscript. We have made all the modifications you requested.

  • Try to change the title to ... during "postharvest life"
  • Answer: - done, (Influence of Wax and Silver Nanoparticles on Preservation Quality of Murcott Mandarin Fruit During Postharvest Life).
  • Please rewrite purpose of your study, focusing on Mandarin no all of fruits.
  • Answer: - done.
  • What is fadan?
  • Answer: - A feddan is a unit of areaused in Egypt, but it has been converted to square kilometers (Km2).
  • Please cite those few related published recent references.
  • Answer: - done.
  • Treatments in Table 1 need to change. Please delete " and packaging in PPE".
  • Answer: - done.
  • Please rewrite conclusions for practical application and do not repeat the main results in the conclusion.
  • Answer: - done.

Round 2

Reviewer 1 Report

The authors have addressed the issues pointed out in the last revision adequately. 

Author Response

Thank you, dear Reviewer, for cooperating with us to revise my manuscript and improving its quality.

Reviewer 2 Report

I thank the Authors for making corrections to the manuscript. Please add information in the "statistical analysis" subchapter that before using parametric ANOVA: 1) the compliance of the data population with the theoretical normal distribution was examined (what test, what result) and the homogeneity of variance in the samples was examined (what test, what result). These are the prerequisites for using parametric tests.

Author Response

Thank you, dear Reviewer, for cooperating with us to revise my manuscript and improving its quality.

I thank the Authors for making corrections to the manuscript. Please add information in the "statistical analysis" subchapter that before using parametric ANOVA: 1) the compliance of the data population with the theoretical normal distribution was examined (what test, what result), and the homogeneity of variance in the samples was examined (what test, what result). These are the prerequisites for using parametric tests.

  • Answer: - Before running a one-way ANOVA, pretests were conducted. We tested the normality assumption on sample distributions and obtained p-values of 0.0001. for homogeneity, we used the Levene test with p- value=0.01598

Reviewer 3 Report

Thanks for your revised manuscript. Some required corrections mentioned as follow:

Line 161: TSS percentage

Line 163: TA percentage

Tables 1, 2, 3, 4, 5, 6 and 7: Wax, not wax + 100 ppm nanosilver.

Table 1: 6 days shelf life, not 6 days slife (P)

Figure 1: Caption is not correct.

Author Response

Thank you, dear Reviewer, for cooperating with us to revise my manuscript and improving its quality.

Line 161: TSS percentage

Answer: - done

Line 163: TA percentage

Answer: - done

Tables 1, 2, 3, 4, 5, 6, and 7: Wax, not wax + 100 ppm nanosilver.

Answer: - done

Table 1: 6 days shelf life, not 6 days slife (P)

Answer: - done

Figure 1: The caption is not correct.

Answer: - done; the caption has been modified to clarify the meaning of the experiment